# E proteins control the development of NKγδT cells through their invariant T cell receptor

Ariana Mihai [1,5], Sang-Yun Lee[2,5], Susan Shinton [2], Mitchell I. Parker[3], Alejandra V. Contreras[2], Baojun Zhang [1], Michele Rhodes [2], Roland L. Dunbrack [3], Juan-Carlos Zúñiga-Pflücker [4], Maria Ciofani [1], Yuan Zhuang[1] & David L. Wiest [2] ✉

T cell receptor (TCR) signaling regulates important developmental transitions, partly through induction of the E protein antagonist, Id3. Although normal γδ T cell development depends on Id3, Id3 deficiency produces different phenotypes in distinct γδ T cell subsets. Here, we show that Id3 deficiency impairs development of the Vγ3+ subset, while markedly enhancing development of NKγδT cells expressing the invariant Vγ1Vδ6.3 TCR. These effects result from Id3 regulating both the generation of the Vγ1Vδ6.3 TCR and its capacity to support development. Indeed, the *Trav15* segment, which encodes the Vδ6.3 TCR subunit, is directly bound by E proteins that control its expression. Once expressed, the Vγ1Vδ6.3 TCR specifies the innate-like NKγδT cell fate, even in progenitors beyond the normally permissive perinatal window, and this is enhanced by Id3-deficiency. These data indicate that the paradoxical behavior of NKγδT cells in Id3-deficient mice is determined by its stereotypic Vγ1Vδ6.3 TCR complex.

T cells comprise two major lineages, αβ or γδ, broadly defined by the T cell antigen receptor (TCR) complex they express. These lineages arise from a common progenitor pool[1–3], and their divergence occurs in response to differences in TCR signaling, with weak TCR signals, typically transduced by the pre-TCR complex specifying the αβ T cell fate, and stronger or more prolonged TCR signals emanating from the γδTCR and specifying the γδ T cell fate[4–6]. The TCR signals of differing strength that instruct divergence of the αβ and γδ T lineages depend on the kinases Lck and ERK, which induce transcription factors (TF) of the Egr family, and result in the proportional induction of the helix-loop-helix factor, Id3[6–8]. Id3 is a critical regulator of αβ/γδ T lineage commitment, and it functions by inhibiting DNA binding by a family of TFs termed E box DNA binding proteins (E proteins). E proteins are class-I basic helix-loop-helix (bHLH) family TFs that bind as homo- or heterodimers to CANNTG motifs (E-box sites)[9] and regulate lymphoid

development and function[9–15]. The differences in TCR signal strength that control lineage fate do so by proportional induction of Id3, which produces graded reductions in E protein activity, with weak signals specifying the αβ fate through modest reduction in E protein activity and strong signals causing the profound repression of E protein activity required for γδ T lineage commitment[8,16–18].

Importantly, Id3-deficiency was found to attenuate development of γδ T cells in a defined γδ TCR transgenic model system, supporting the model that TCR signal strength instructs the γδ T cell fate through graded repression of E protein function[8]. However, several groups identified a subset of γδ T cells, characterized by expression of a Vγ1Vδ6 TCR, that was markedly expanded in non-Tg Id3-deficient mice[8,19–21], raising the question of why Id3-deficiency did not attenuate their development. A distinguishing aspect of γδ T lymphocyte ontogeny is the ordered appearance of waves of γδ T cells defined by their

[1]Immunology Department, Duke University, Durham, NC, USA. [2]Nuclear Dynamics and Cancer Program, Fox Chase Cancer Center, Philadelphia, PA, USA. [3]Cancer Signaling and Microenvironment Program, Fox Chase Cancer Center, Philadelphia, PA, USA. [4]Immunology Department, University of Toronto, Toronto, CA, USA. [5]These authors contributed equally: Ariana Mihai, Sang-Yun Lee. ✉e-mail: David.Wiest@fccc.edu

Vγ usage[22,23], with each wave associated with an effector function and anatomic location[24]. The first such wave, peaks around embryonic day 14 (E14) and comprises Vγ3+ dendritic epidermal T cells (DETCs), which produce interferon-γ (IFNγ) and home to the epidermis[25]. The next waves occur at E16 and E18 and comprise Vγ4+ and Vγ2+ subsets, respectively, which are associated with IL-17 production and residence in the lung, uterus, and lymphoid tissues[24,26,27]. The final wave begins during late fetal life and comprises Vγ1+ T lymphocytes, among which is the natural killer γδ T cell (NKγδT) subset[27–30]. NKγδT cells are characterized by a stereotypic Vγ1.Vδ6.3 TCR, PLZF expression, and co-production of IL-4 and IFNγ without the need for prior activation[19,31,32]; and their development is restricted to the late fetal/perinatal developmental window[19,31,32].

Development of PLZF-expressing NKγδT cells relies on strong TCR signaling as well as the Signaling Lymphocyte Adaptor Molecule (SLAM)/SLAM-Associated Adaptor Protein (SAP) pathway, which induces *Zbtb16*, *Zbtb7b*, and *Klf2*, and confers upon NKγδT cells their characteristic phenotype and cytokine secretion profile[19,33–35]. Vγ1.Vδ6.3-expressing NKγδT cells expand dramatically in mice lacking downstream factors of TCR-signaling such as Itk, SLP-76, and Id3[19,21,36–40]. This suggests strong TCR signals emanating from the Vγ1.Vδ6.3 TCR, likely in response to engagement by a high affinity ligand, normally lead to deletion; however, in the context of settings where TCR-signaling is attenuated by the absence of key regulators, these cells escape from negative selection and expand[8]. This interpretation was borne out by a direct test using a high affinity ligand for a defined γδ TCR, which resulted in deletion of γδ T cells that were then rescued by Id3-defiiency[8]. Nevertheless, the mechanistic basis by which Id3-defiency causes expansion of Vγ1.Vδ6.3 NKγδT cells remains unclear.

Here we explore the mechanistic basis by which disturbing the genomic landscape by Id3-deficiency results in expansion of the Vγ1.Vδ6.3 NKγδT cells. Indeed, we determined that the *Trav15* segment, which encodes the Vδ6.3 TCR subunit, is an E protein target whose incorporation into the Vγ1Vδ6.3 TCR is regulated by E protein binding and by generally altering the E protein activity through Id3-deficiency. Moreover, once expressed, the resulting the Vγ1Vδ6.3 TCR clonotypes that expand in Id3-deficient mice also appear to more effectively specify the innate-like NKγδT cell fate in a cell-autonomous and developmentally-unrestricted manner, providing an explanation for their expansion in Id3-deficient mice. Together, these data indicate that the ability of NKγδT cells to markedly expand in the Id3-deficient setting is entirely determined by its stereotypic Vγ1Vδ6.3 TCR complex.

## Results
### Id3 loss has selective effects on Vγ subsets
TCR signal strength/duration plays a key role in separation of the αβ and γδ lineages through induction of the HLH factor, Id3[4,6,8,41–44]. Interestingly, however, we, and others, have previously reported the paradoxical finding that loss of the Id3 antagonist results in the marked expansion of Vγ1Vδ6.3 NKγδT cells[8,19,21,36]. This was not observed in E19.5 fetal thymus (Supplementary Fig. 1a, b), but was quite profound in Id3-deficient adults, which exhibited a marked expansion of Vγ1+ γδ T cells (Fig. 1a–d). Nevertheless, development of the Vγ2+ and skin Vγ3+ DETC subsets was impaired by Id3-deficiency (Fig. 1a–f). To determine if the reduction of Vγ3 DETC γδ T cells in the skin of Id3-deficient mice (Fig. 1e, f) resulted from impaired development in the thymus, we performed fetal thymic organ culture (FTOC) analysis on fetal thymic lobes from Id3+/+, Id3+/-, and Id3-/- mice (Fig. 2). While the development of CD4+CD8+ (DP) αβ T lineage cells was not impaired, development of Vγ3+ DETC γδ T cells was impaired by Id3-deficiency (Fig. 2a, b). FTOC analysis of Id3-/- thymic lobes revealed that Vγ3+ thymocytes were equally represented early in development (E15.5), but failed to be selected and expand over time, as indicated by the failure to induce

CD122 expression (Fig. 2b, c)[43]. Moreover, Id3-deficient Vγ3+ progenitors exhibited a clear increase in generation of CD4+CD8+ (DP) thymocytes, suggesting that the loss of Id3 results in diversion of some cells to the αβ T cell fate (Supplementary Fig. 1c, d), as we have reported previously[8]. Because the γδ TCR is downmodulated during development to the DP stage, this presumably represents an underestimate of the extent of diversion to the αβ T cell fate (Supplementary Fig. 1c, d)[6,44].

### The Vγ1Vδ6.3 subset that expands in Id3-deficient mice exhibits CDR3 restriction
The basis for expansion of the Vγ1Vδ6.3 subset in Id3-deficient mice remains unexplained. Our previous analysis[8] raised the possibility that a selected subset of Vγ1Vδ6.3 TCRs, possibly with high affinity for ligand, were expanded in Id3-deficient mice. To test this possibility, we isolated immature (CD24hi) and mature (CD24low) thymic Vγ1Vδ6.3 γδ T cells from Id3+/+ and Id3-/- mice, and sequenced their TCR γ and δ chains (Fig. 3a and Supplementary data 1). This revealed that the CDR3 γ and δ sequences of the Vγ1Vδ6.3 TCRs were quite diverse in immature CD24hi progenitors of both Id3+/+ and Id3-/- mice; however, upon maturation into CD24low cells, the CDR3γ and δ sequences of Vγ1Vδ6.3 cells from Id3-/- mice became less diverse and shorter in length, than those from Id3+/+ mice (Fig. 3b). The WT structures exhibited much lower pLDDT values and much greater structural variability than the KO structures. This may be because the WT sequence modeled was longer (21 residues) than the KO sequence (17 residues)(Fig. 3c), as is generally true, since WT sequences are longer and more diverse (Fig. 3b). Structural modeling of the CDR3s indicated that those from Id3-/- mice exhibited less flexibility and freedom of movement (Fig. 3c). The marked restriction of the CDR3 sequences in the Vγ1Vδ6.3 TCRs of mature γδ T cells from Id3-/- mice could have resulted either from a perturbation of terminal deoxynucleotide transferase (TDT) expression impacting N-region addition and/or from ligand-mediated selection of these cells.

### Vγ1Vδ6.3 TCR is sufficient to instruct the NKγδT cell fate and promote clonal dominance in Id3-deficient mice
To determine if the Vγ1Vδ6.3 TCRs that expand in Id3-deficient mice have the capacity to autonomously promote clonal dominance, we generated transgenic (Tg) mice with inducible expression of the Vγ1Vδ6.3 TCR. To do so, we employed one of the Vγ1Vδ6.3 TCRs that expanded among mature CD24low thymocytes in Id3-deficient mice and compared it to a Vγ1Vδ6.3 TCR from Id3-sufficient mice (Fig. 4a). The Vγ1Vδ6.3 TCR Tg were knocked into the *Rosa26* locus immediately downstream of a lox-stop-lox (LSL) cassette that represses expression until Cre-mediated excision of the LSL (Fig. 4a). The TCRγ and TCRδ subunits are translated from the same transcript and the fusion protein is joined by a Tescovirus 2 A (2 A) sequence that enables stoichiometric expression of the separable TCRγ and TCRδ subunits (Fig. 4a). To assess the capacity of the Vγ1Vδ6.3 TCR Tg to exhibit clonal dominance, the LSL was selectively excised using 4-hydroxytamoxifen induction of *Tcrd-Cre*, which is active in a subset of T cell progenitors[45,46], enabling an assessment of the ability of the induced TCR to exert clonal dominance (Fig. 4b). Induction of the Vγ1Vδ6.3 TCRs derived from Id3+/+ (WT) and Id3-/- (KO) mice in an Id3-sufficient background (Id3+/-), resulted in modest development of PLZF-expressing Vγ1Vδ6.3 NKγδT cells, with both the WT and KO TCRs exhibiting similar ability to instruct the NKγδT cell fate (Fig. 4c, d). However, in an Id3-deficient host (Id3-/-), both TCR complexes exhibited a significantly-enhanced capacity to promote clonal dominance, as the frequency of TCRδ+ cells and the absolute number of cells expressing the TCR Tg, and the lineage-defining TF PLZF, was markedly increased (Fig. 4c, d). The capacity of the WT TCR to do so is somewhat surprising given its lack of predominance among the expanded clonotypes observed among mature NKγδT in Id3-deficient

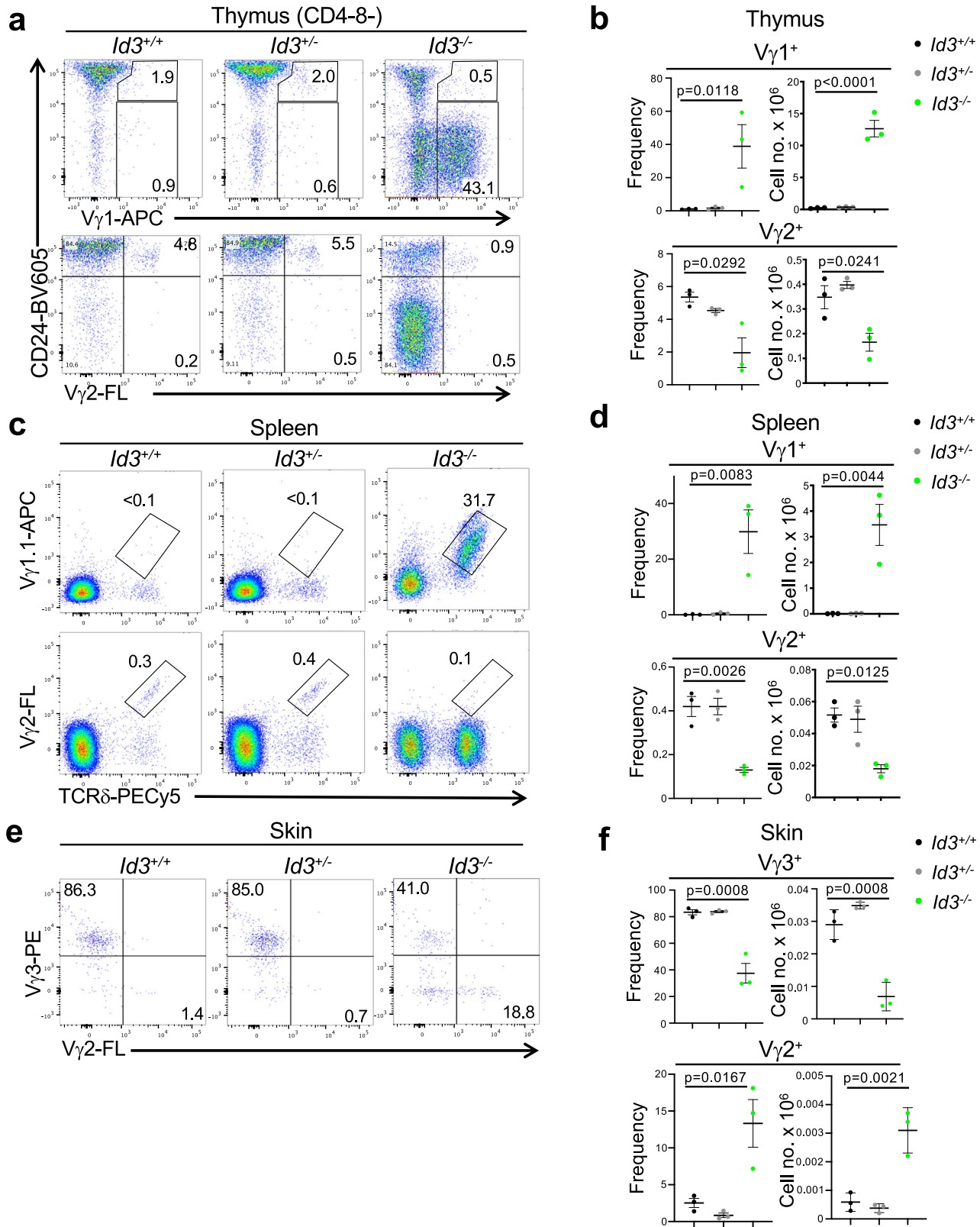

mice (Supplementary data 1). Nevertheless, consistent with its dominance in *Id3*[-/-] mice, the KO TCR exhibited a significantly greater capacity than the WT TCR to promote adoption of the NKγδT fate in *Id3*[-/-] mice (Fig. 4c, d). In parallel analysis, conditional ablation of *Id3*[fl/fl] alleles using *Tcrd-Cre*[ER] also revealed that the KO TCR led to more extensive expansion of NKγδT cells than did that WT TCR, further supporting the interpretation that the Vγ1Vδ6.3 TCRs are capable of

promoting NKγδT lineage specification in a cell autonomous manner (Supplementary Fig. 2). Taken together, these data suggest that the Vγ1Vδ6.3 TCR clonotypes that preferentially expand among NKγδT cells in Id3-deficient mice are also more capable of promoting NKγδT cell development in Id3-deficient progenitors in this acute assay, even when expressed beyond the fetal/perinatal window to which their development is normally restricted[27].

**Fig. 1 | Impact of Id3-deficiency on development of γδ T cell subsets.**
**a, b** Representative flow cytometry plots of thymocytes from Id3⁺/⁺, Id3⁺/⁻, and Id3⁻/⁻ mice. **a** Flow cytometry plots of expression of CD24 and the indicated Vγ are displayed for CD4⁻CD8⁻ thymocytes. **b** A graphical representation is depicted of the frequency and mean absolute number ± S.D. of Vγ2⁺ and Vγ1.1⁺ thymocytes from Id3⁺/⁺, Id3⁺/⁻, and Id3⁻/⁻ mice calculated from gate frequencies. **c** Representative flow cytometry plots displaying staining for Vγ1.1 vs TCRδ, and Vγ2 vs TCRδ on splenocytes from Id3⁺/⁺, Id3⁺/⁻, and Id3⁻/⁻ mice. **d** A graphical representation is depicted of the frequency and mean absolute number ± S.D. of Vγ2⁺ and Vγ1.1⁺ splenocytes

from Id3⁺/⁺, Id3⁺/⁻, and Id3⁻/⁻ mice calculated from gate frequencies. **e** Representative flow cytometry plots of Vγ2⁺ and Vγ3⁺ staining on skin preparations from Id3⁺/⁺, Id3⁺/⁻, and Id3⁻/⁻ mice. **f** A graphical representation is depicted of the frequency and mean absolute number ± S.D. of Vγ2⁺ and Vγ3⁺ skin γδ T cells from Id3⁺/⁺, Id3⁺/⁻, and Id3⁻/⁻ mice calculated from gate frequencies is depicted. 3 C57BL/6 background mice of the indicated genotypes were analyzed in the depicted experiment. Data are representative of 3 independent experiments. Statistical analysis: one-way ANOVA with correction for multiple comparison using Tukey's post hoc test.

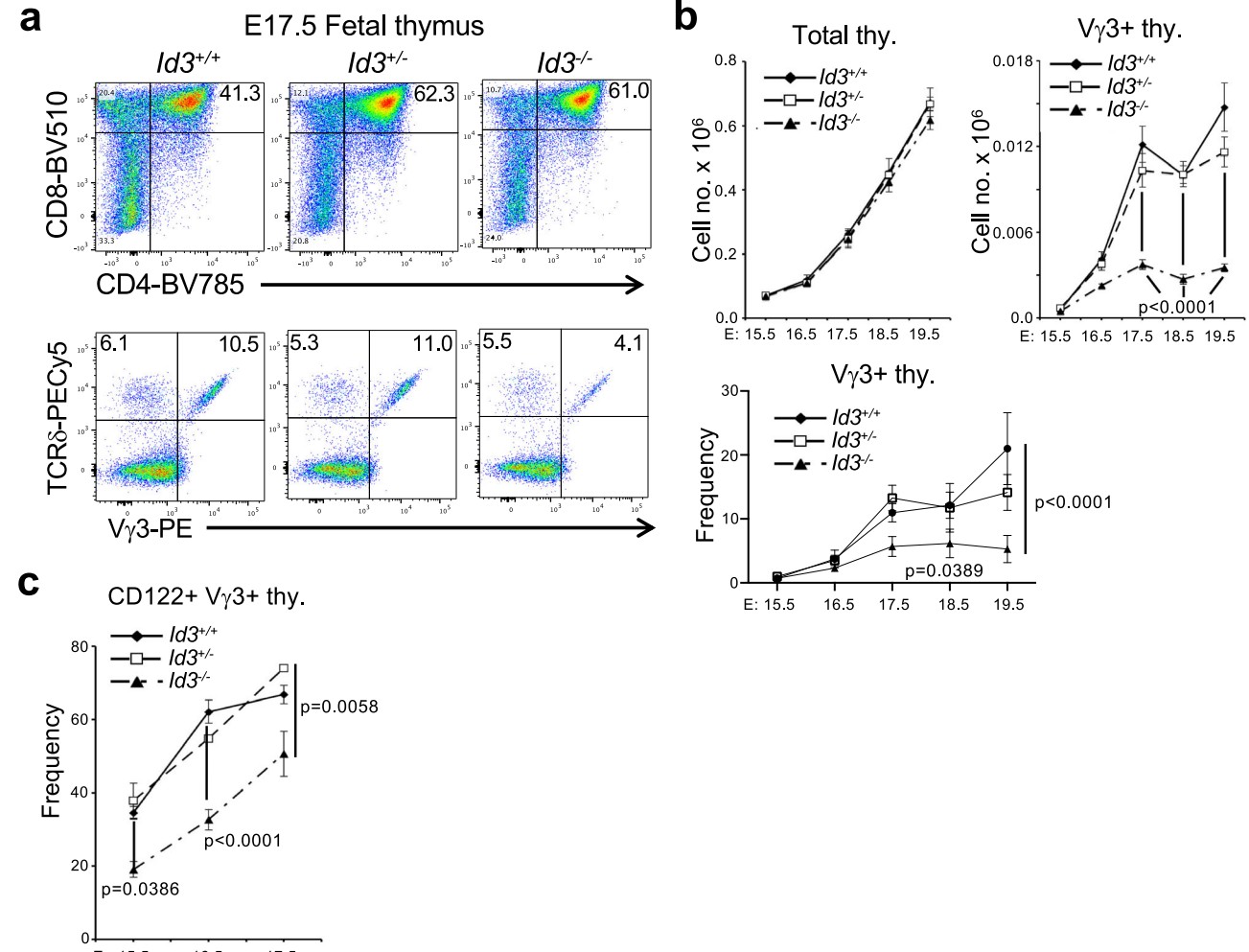

**Fig. 2 | Effect of Id3-deficiency on fetal development of Vγ3 + γδ T cells.**
**a** Representative flow cytometry plots of CD4/CD8 and TCRδ/Vγ3 staining of E17.5 thymocytes from Id3⁺/⁺, Id3⁺/⁻, and Id3⁻/⁻ C57BL/6 background mice. **b** Graphic depiction of the mean absolute number ± S.D. of total thymocytes per lobe (left) or frequency (bottom) and absolute number (right) Vγ3⁺ γδ progenitors at E15.5 and at

each of 4 additional days in fetal thymic organ culture (FTOC: right). **c** Mean frequency ± S.D. of the fraction of Vγ3⁺ γδ progenitors expressing CD122 is displayed graphically. 4 thymic lobes per genotype were analyzed at each time point. Statistical analysis: two-way ANOVA with correction for multiple comparison using Tukey's post hoc test.

The KO Vγ1Vδ6.3 TCR also exhibits a somewhat greater capacity to promote the development of Id3⁺/⁺ progenitors into PLZF-expressing NKγδT cells in vitro. Cre-mediated induction of the Vγ1Vδ6.3 TCR Tg in Id3⁺/⁺ fetal liver (FL) precursors cultured on OP9-DL1 resulted in induction of PLZF expression by progenitors expressing both the WT and KO Vγ1Vδ6.3 TCR (Supplementary Fig. 3a–d). Moreover, the KO Vγ1Vδ6.3 TCR appeared to promote somewhat more extensive differentiation of mature CD73⁺CD24ᴸᵒʷ cells that express PLZF (Fig. 3a–d). Because NKγδT cell development is typically restricted to the late fetal/perinatal developmental window, we asked whether the KO Vγ1Vδ6.3 TCR was able to instruct development of

NKγδT cells beyond the fetal/perinatal window. To test this possibility, we induced the KO Vγ1Vδ6.3 TCR in adult bone marrow progenitors and found it was able to instruct the NKγδT cell fate. Importantly, FL precursors exhibited a greater capacity to support Vγ1Vδ6.3 TCR-mediated induction of CD73 and PLZF expression than did the bone marrow progenitors (Supplementary Fig. 3e–g); however, enforced expression of the Vγ1Vδ6.3 TCR in adult bone marrow progenitors was also able to promote NKγδT cell development. This suggests that once expressed, the Vγ1Vδ6.3 TCR complexes specify the NKγδT cell fate in a cell-autonomous, developmentally-unrestricted manner. Finally, while the greater permissiveness of fetal precursors to adoption of the

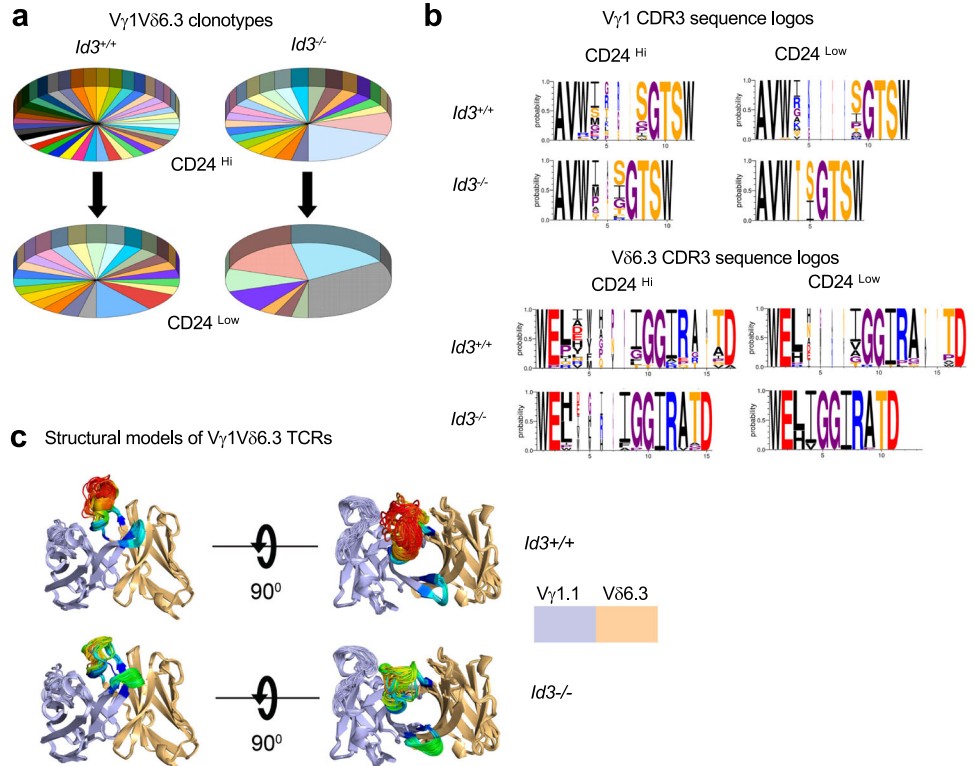

**Fig. 3 | Alterations in Vγ1.1Vδ6.3 clonotypes in Id3-deficient mice. a** Pie charts depicting the frequencies of Vγ1.1Vδ6.3 CDR3 clonotypes identified by single-cell TCR sequencing in CD24hi and CD24low Vγ1.1Vδ6.3 thymocytes from Id3+/+ and Id3−/− mice. Id3+/+ CD24hi, n = 40; Id3+/+ CD24low, n = 37; Id3−/− CD24hi, n = 30; Id3−/− CD24low n = 30; **b** Logos depicting the length and diversity of CDR3 sequences from the Vγ1.1 and Vδ6.3 subunits of CD24hi and CD24low Vγ1.1Vδ6.3 thymocytes from Id3+/+ and Id3−/− mice. **c** Structural models of CDR3 sequences from the Vγ1.1 and Vδ6.3 subunits of CD24low Vγ1.1Vδ6.3 thymocytes from Id3+/+ and Id3−/− mice, depicting their predicted freedom of movement. Structure alignment of AlphaFold-Multimer v2.3 models of a single KO and a single WT γδ pair, chosen as representative of the sequence distribution. The frameworks and CDRs 1 and 2 are colored by TCG gene (delta=blue, gamma=orange). The CDRs are colored according to AlphaFold's pLDDT scores, which are a measure of the predicted accuracy of the environment of each residue[71]. The coloring ranges from blue (most confident) to red (least confident).

NKγδT cell fate suggests that cellular context may influence the effectiveness with which the Vγ1Vδ6.3 TCR can instruct NKγδT cell development (Supplementary Fig. 3e−g), these data indicate that restriction of NKγδT cell development to the fetal/neonatal gestational period may result from selective generation of the Vγ1Vδ6.3 TCR, perhaps under the control of the Id3/E protein axis.

### Vδ gene segment utilized to generate the Vγ1Vδ6.3 TCR
Since Vγ1+ γδ T cells that are Vδ6.3− continue to be generated beyond the late fetal/neonatal period, we reasoned that the restricted developmental timing of NKγδT cells may be in part regulated at the level of Vδ6.3 rearrangement. NKγδT cells have been reported to have nonfunctional Tcrd rearrangements utilizing a variety of Vδ gene segments, indicating their precursors are capable of rearranging any Vδ gene segment[47]. Nevertheless, the TCRδ chain utilized by NKγδT thymocytes results almost exclusively from rearrangement of the Trav15d-1-dv6d-1 gene segment (Vδ6.3 TCR chain), with a minority employing the Trav15-1-dv6-1 gene segment (Vδ6.1 TCR chain)[47]. To determine if the Id3/E axis influenced the utilization of those Trav15 family V gene segments, E protein binding at the Tcra-Tcrd locus was examined by performing E2A and HEB ChIP-seq of FL derived RAG2-deficient CD4−CD8−CD44−CD25+ (DN3) T cell precursors (Supplementary Fig. 4a)[48]. Interestingly, of all the Vα/Vδ segments, Trav15d-1-dv6d-1, Trav15n-1, and Trav15-1-dv6-1 showed substantial E2A and HEB binding, which was present in both the promoter region and downstream of the RSS (Fig. 5a−c). This suggests that Tcrd recombination of Trav15-dv6 family V gene segments is regulated in cis by E proteins, likely by promoting chromatin accessibility. Consequently, it is possible that

Id3-deficiency could influence utilization of these segments through enhancing E protein binding.

To identify the Trav15 element required to support development of NKγδT cells, we targeted them for deletion by CRISPR/Cas9 in TcrdCreER mice, which lack the Trav15n-1 segment[45]. The individual deletions of Trav15d-1-dv6d-1 (Δ15d-1) and Trav15-1-dv6-1 (Δ15-1) spanned ~2.1 kb over the respective gene segment and eliminated the E protein binding regions both in the promoter and downstream of the RSS (Fig. 5d). We additionally produced a Tcra-Tcrd allele with a 430 kb deletion within the Vα/Vδ array (Δ430), which eliminated both the 15d-1 and 15-1 elements, as well as the intervening 430 kb with the V gene segments contained therein (Fig. 5d). None of the resulting mutant mice exhibited alterations in thymic cellularity, CD4/8 subsets, or total γδ T cells (Fig. 5e and Supplementary Fig. 4b, c). As expected, ablation of both Trav15-1 family gene segments (Δ430) blocked development of NKγδT cells (Fig. 5e). Importantly, while ablation of the proximal Trav15 segment, Trav15-1-dv6-1 (Δ15-1), failed to block development of NKγδT cells, their development was severely attenuated by ablation of the distal segment, Trav15d-1-dv6d-1 (Δ15d-1) (Fig. 5e), demonstrating that NKγδT cell development depends predominantly on use of Trav15d-1-dv6d-1 (Fig. 5e).

We next sought to determine whether globally-enhancing E protein activity through Id3-deficiency would alter the capacity of Trav15 family members to support expansion of γδ T cells expressing the Vγ1Vδ6.3 TCR. Id3-deficiency in TcraΔ430/Δ430 mice did not lead to observable differences in thymic cellularity or γδ T lymphocytes (Fig. 5f and Supplementary Fig. 4d); however, elimination of Id3 in TcraΔ15-1/Δ15-1 mice resulted in a significant reduction of thymic cellularity

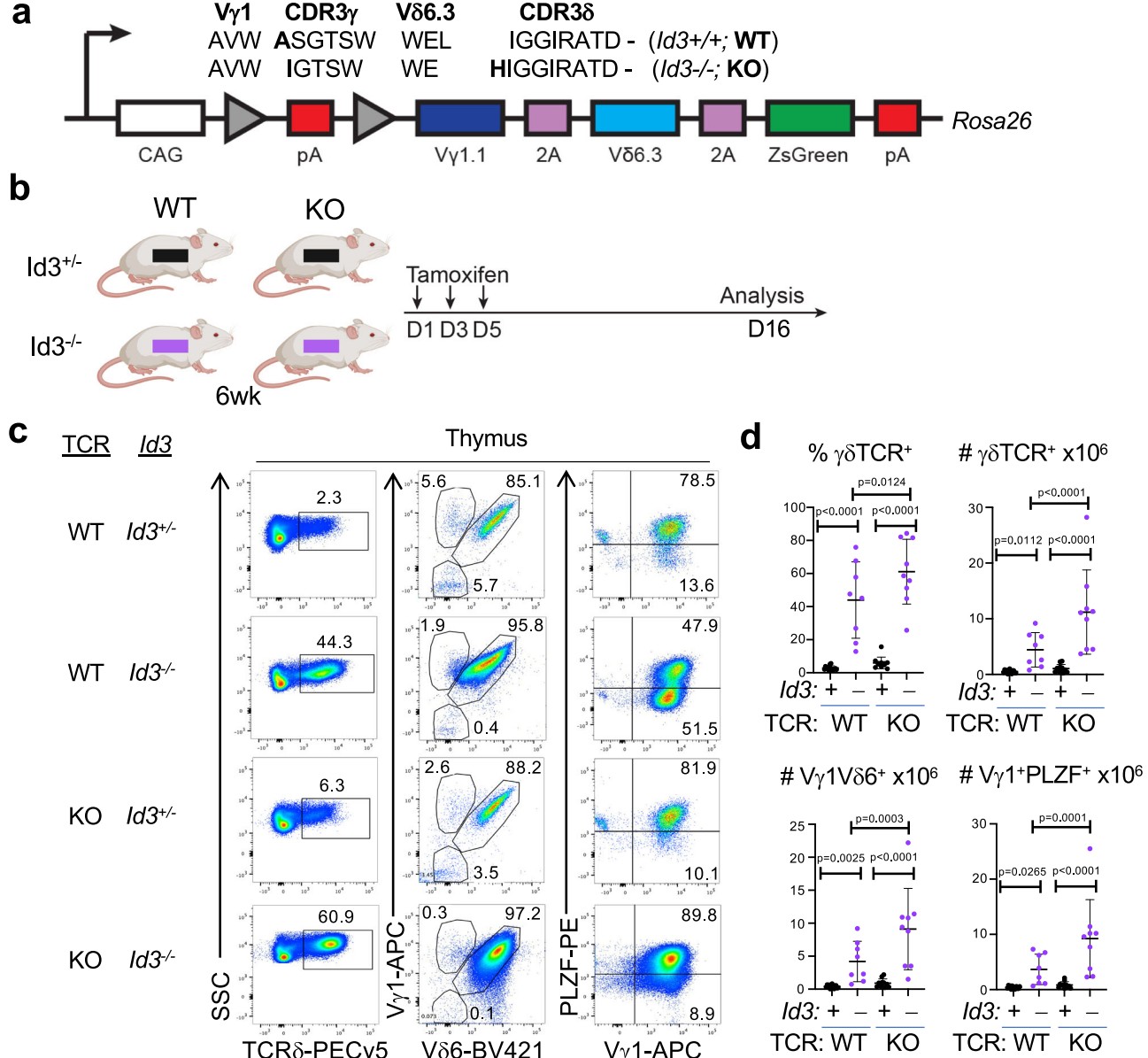

**Fig. 4 | Capacity of Vγ1.1Vδ6.3 Tg from Id3⁺/⁺ and Id3⁻/⁻ mice to support development of NKγδT cells. a** Diagram of *Rosa26* targeted Vγ1.1Vδ6.3 TCR transgene. CAG transcription is blocked by a polyadenylation (pA) sequence in the LSL element. The Vγ1.1 and Vδ6.3 TCR chains are linked by a self-cleaving Tescovirus P2A sequence. ZsGreen fluorescent reporter is linked to the TCR construct by a self-cleaving T2A sequence. The amino acid sequence comparison of the generated TCR transgenes is shown. Sequence differences are in bold. **b** Diagram of in vivo experimental design, constructed using BioRender. Tamoxifen was administered to 6-week-old mice on days 1, 3, and 5, and analysis was performed on day 16.

**c** Representative flow cytometry plots for ZsGreen⁺ cells stained with the indicated antibodies. Frequency of γδ T lymphocytes, and absolute numbers of γδ T cells, Vγ1.1Vδ6.3 expressing cells, and Vγ1⁺PLZF⁺ γδ T cells in Id3⁺/⁻ and Id3⁻/⁻ mice are depicted graphically (**d**) as scatter grams. Each dot represents an individual mouse and statistical significance was determined using two-way ANOVA with correction for multiple comparison using Tukey's post hoc testing. Mean ± S.D. of the measurements is overlayed on the scatter grams. WT TCR Id3⁺/⁻, *n* = 14; WT TCR Id3⁻/⁻, *n* = 8; KO TCR Id3⁺/⁻, *n* = 11; KO TCR Id3⁻/⁻, *n* = 9.

and a significant expansion of Vγ1.1⁺ T lymphocytes (Fig. 5f and Supplementary Fig. 4d), consistent with our finding that NKγδT cell development depends primarily on *Trav15d-1-dv6d-1*, which is intact in these mice. Interestingly, Id3-deletion in Tcra^Δ15d-1/Δ15d-1 mice also resulted in a significant reduction of thymic cellularity and a significant expansion of Vγ1.1⁺ T lymphocytes which were largely PLZF⁺ (Fig. 5f and Supplementary Fig. 4d, e) and thus appear to be NKγδT cells whose expansion was supported by the Vγ1.1Vδ6.1 TCR. Taken together, these data indicate that development of NKγδT is primarily dependent upon the *Trav15d-1-dv6d-1* element for encoding the TCRδ subunit of their stereotyped Vγ1.1Vδ6.3 TCR complex; however, when E protein activity is enhanced by Id3-deficiency and the *Trav15d-1-*

*dv6d-1* element is eliminated, then the *Trav15-1-dv6-1* element is capable of supporting NKγδT cell development through a Vγ1.1Vδ6.1 TCR complex.

**E protein bound elements supporting NKγδT cell development**
Having determined that generally enhancing E protein activity altered the usage of *Trav15* gene segments by developing NKγδT cells, we wished to assess the role of cis-acting E protein binding sites adjacent to the *Trav15* element (*Trav15d-1-dv6d-1*) preferentially utilized by Vγ1.1Vδ6.3 expressing NKγδT cells. There are three consensus E box binding sites within the first 100 bp downstream of the RSS of the *Trav15d-1-dv6d-1* gene segment (Fig. 6a). To assess their role in

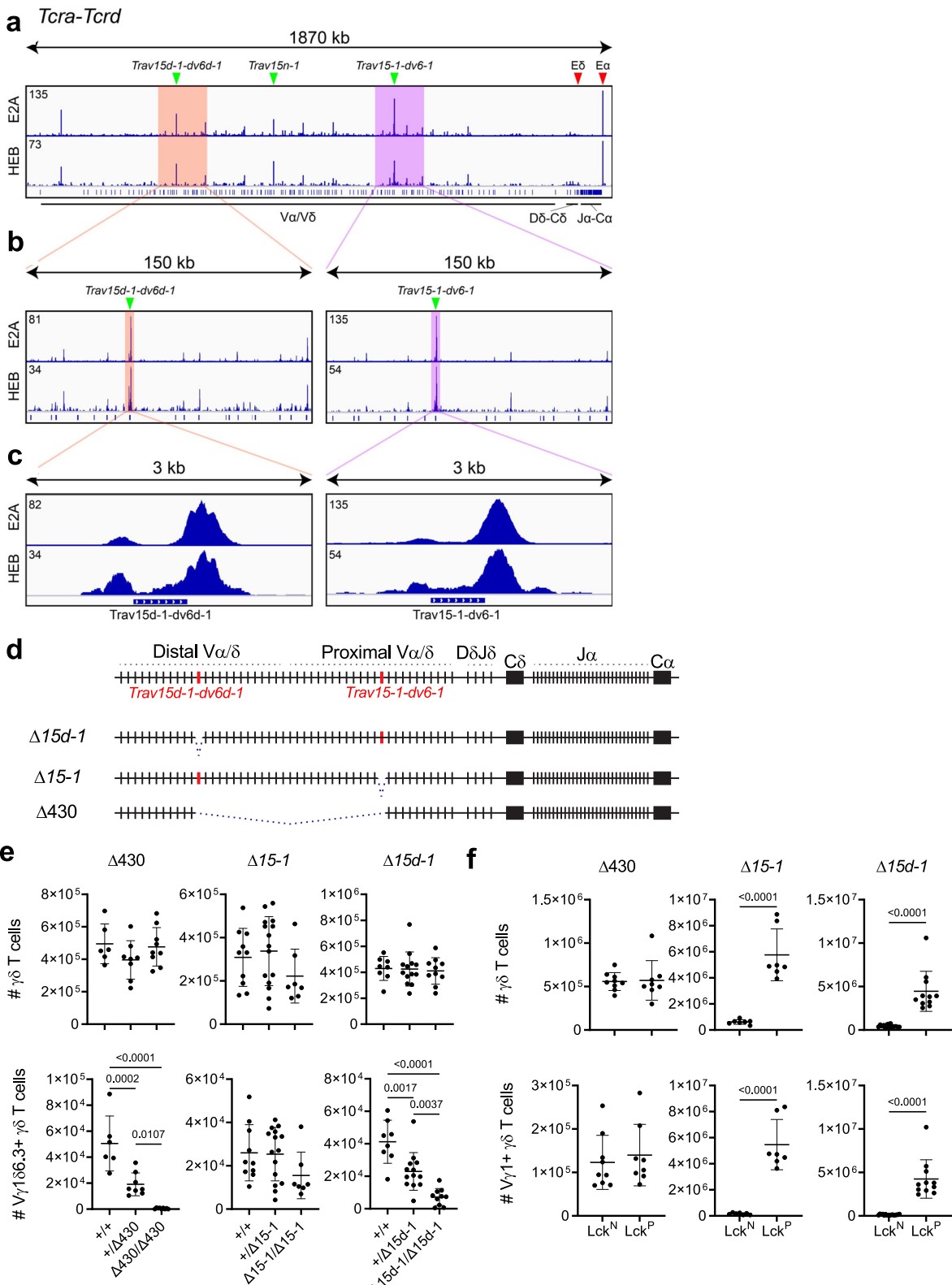

supporting the *Trav15d-1-dv6d-1*-encoded Vδ6.3 subunit of the TCR that supports NKγδT cell development, the E-box containing region downstream of *Trav15d-1-dv6d-1* was ablated by CRISPR editing (Fig. 6a; *ΔE*). It should be noted that to eliminate the possibility of compensation by the *Trav15-1-dv6-1* element, we ablated the E protein binding sites on the *Δ15-1* allele described above, which lacks the *Trav15-1-dv6-1* element. The *ΔE* mutation did not affect thymic

cellularity or the total number of γδ T cells (Fig. 6b and Supplementary Fig. 4f); however, it did reduce both the frequency and number of Vγ1Vδ6.3 expressing NKγδT cells (Fig. 6b), suggesting that E protein binding downstream of *Trav15d-1-dv6d-1* element plays a role in regulating recombination at this *Trav15* element and supporting the expression of the Vγ1Vδ6.3 TCR preferred by NKγδT cells. Because predicted binding sites for other TFs mapped to the 59 bp ablated in

**Fig. 5 | *Trav15* family members supporting development of NKγδT cells. a** ChIP-seq analysis of E2A and HEB binding to the C57BL/6 *Tcra-Tcrd* locus with positions of Eδ and Eα indicated by red arrows, and *Trav15d-1-dv6d-1*, *Trav15n-1*, and *Trav15-1-dv6-1* indicated by green arrowheads. Progressively zoomed in view of the 150 kb region (**b**) or 3 kb region (**c**) highlighting the region of the *Tcra-Tcrd* locus corresponding to the *Trav15d-1-dv6d-1* and *Trav15-1-dv6-1* elements (green arrowheads). Blue bar with rightward checks indicates the location of the RSS. The maximum ChIP-seq peak value is indicated in the top left corner of the trace. Vertical blue lines in (**a**) denote V, D, and J gene segments, or constant region(s) as indicated. **d** Diagram of *Tcra-Tcrd* locus in 129 strain mice with relative positions of *Trav15d-1-dv6d-1* and *Trav15-1-dv6-1* indicated. Mutant alleles bearing deletions of *Trav15d-1-dv6d-1* (*Δ15d-1*), *Trav15-1-dv6-1* (*Δ15-1*), or the entire 430 kb interval (*Δ430*) are shown below. **e** Number of thymic TCRδ⁺ and Vγ1.1⁺Vδ6.3⁺ cells summarized from

analysis of *Δ430*, *Δ15-1*, and *Δ15d-1* mutants. Littermates were used for all comparisons. *Δ15-1*: *Tcra⁺/⁺* (*n* = 10), *Tcra⁺/Δ15-1* (*n* = 15), and *Tcra^Δ15-1/Δ15-1* (*n* = 7); *Δ15d-1*: *Tcra⁺/⁺* (*n* = 8), *Tcra⁺/Δ15d-1* (*n* = 13), and *Tcra^Δ15d-1/Δ15d-1* (*n* = 10); *Δ430*: *Tcra⁺/⁺* (*n* = 6), *Tcra⁺/Δ430* (*n* = 8), and *Tcra^Δ430/Δ430* (n = 9) Data are plotted as mean ± SD and are pooled from at least 3 independent experiments. Statistical analysis: one-way ANOVA with correction for multiple comparison using Tukey's post hoc testing. **f** The number of thymic TCRδ⁺ and Vγ1.1⁺Vδ6.3⁺ cells in Lck-Cre negative (Lck^N) Id3 sufficient (*Id3^fl/fl*) (control), or Lck-Cre (Lck^P) mediated Id3 deficient (*Id3^fl/fl*) *Δ430*, *Δ15-1*, and *Δ15d-1* mutants. All comparisons to Lck-Cre negative (Lck^N) littermates. For *Δ430*: Lck^N (*n* = 9), Lck^P (*n* = 8); *Δ15-1*: Lck^N (*n* = 7), Lck^P (*n* = 7); *Δ15d-1*: Lck^N (*n* = 16) and Lck^P (*n* = 11). Data are pooled from at least 3 independent experiments and plotted as mean ± SD. Statistical analysis: Two-sided student's *t* test.

*ΔE*, such as Sox4 and Runx1, we directly tested the role of the E protein binding sites by selectively inactivating the 1st (*ΔE1*) and 3rd (*ΔE3* and *mE3*) E box sites (Fig. 6a, b). As with deletion of the entire 59 bp interval, inactivation of the 1st or 3rd E box sites did not alter the total number of γδ T cells (Fig. 6b); however, inactivation of those E box sites did reduce both the frequency and number of Vγ1Vδ6.3 expressing NKγδT cells (Fig. 6b).

Taken together, our analysis indicates the Id3/E protein axis plays an important role in both generation of the Vγ1Vδ6.3 TCR complex and its capacity to specify the NKγδT cell fate (Fig. 7).

## Discussion

Previous studies support the hypothesis that TCR signals of differing strength regulate the separation of the αβ and γδ lineage fates through Id3-mediated graded repression of the function of E proteins, with weak signals causing modest Id3 induction, retention of significant E protein activity, and adoption of the αβ fate while strong TCR signals markedly repress E protein activity and promote the γδ fate[8,48,49]. While the central role of Id3 in αβ/γδ lineage commitment is supported by studies with defined γδ TCR complexes[8] and for the Vγ3⁺ DETC cells whose development is attenuated by Id3-deficiency, development of Vγ1Vδ6.3 expressing NKγδT cell stands in stark contrast, as NKγδT cells are not only not dependent on Id3, but are actually restrained by it[8,19–21]. Here we provide an explanation for this conundrum, revealing that the Id3/E protein axis plays a key role in regulating the development of NKγδT cells, by controlling both the generation of the stereotyped NKγδT Vγ1Vδ6.3 TCR complex, and its capacity to support NKγδT cell development (Fig. 7).

Our sequence analysis revealed that the Vγ1.1Vδ6.3 TCR complexes of the NKγδT cells that expanded in Id3-deficient mice were characterized by CDR3 sequences that were shorter and less diverse than those found in Id3-sufficient mice. This may be related to the absence of TDT, which is required for non-templated addition of nucleotides (N) to the CDR3 junctions[50,51]. The Vγ1.1Vδ6.3 TCR complexes of NKγδT cells may lack these N-additions either because TDT is not expressed during fetal gestation when NKγδT cells develop[50], or because E proteins regulate TDT expression, as has been reported in B lineage cells[52]. Importantly, E protein regulation of TDT seems an unlikely explanation, since Id3-deficiency enhances E protein activity, which should increase TDT expression[52] and thus N-region additions. Accordingly, a more likely explanation for the expansion in Id3-deficient mice of NKγδT cells with restricted CDR3s is a change in competitive fitness of progenitors bearing Vγ1.1Vδ6.3 TCR complexes in the context of Id3-deficiency, perhaps in response to ligand-mediated selection. This is consistent with our previous analysis using KN6 γδ TCR Tg mice, which showed that γδ T cell development in response to TCR signals induced by low affinity ligands was blocked by Id3-deficiency, while development in response to TCR signals elicited by high-affinity ligands that normally delete or restrain γδ T cell development in Id3-sufficient mice, promoted expansion in Id3-deficient mice[8]. Likewise, we determined that conditional expression

of Vγ1.1Vδ6.3 TCR complexes in adult Id3-sufficient mice (6wk old) was able to instruct the development of PLZF-expressing NKγδT cells, albeit in limited numbers. Importantly, when the Vγ1.1Vδ6.3 TCR complexes were induced in Id3-deficient mice, development of PLZF-expressing NKγδT cells was markedly enhanced. Both the Vγ1.1Vδ6.3 TCR complexes isolated from Id3-sufficient and Id3-deficient mice were able to promote NKγδT cell development, although the Vγ1.1Vδ6.3 TCR complexes isolated from Id3-defiicient (KO) mice exhibited a significantly greater ability to do so, providing an explanation for its preferential expansion in Id3-deficient mice.

The ability of the Vγ1.1Vδ6.3 TCR complexes to support the profound expansion of NKγδT cells even in the context of Id3-deficiency raises the question of why they support expansion when other γδ TCR complexes (KN6 and Vγ3⁺ DETC) are unable to do so[8]. The restricted length and diversity of the CDR3 sequences of the TCRγ and TCRδ chains likely stems from TCR-ligand mediated selection, as has been previously suggested[8,47]. The putative intrathymic ligand for Vγ1.1Vδ6.3 TCR complexes has not been identified, but it likely involves polyanionic features, as polyanions have been shown to activate Vγ1.1Vδ6.3 TCRs in vitro[53]. The Vγ1.1Vδ6.1 and Vγ1.1Vδ6.3 TCRs likely recognize the same ligand, with the latter possibly having higher binding affinity which could support their ability to outcompete Vγ1.1Vδ6.1 T lymphocytes for intrathymic encounters with a high-affinity ligand. In this scenario, Itk- or Id3-deficiency attenuates the capacity of a strong TCR-ligand interaction to transmit a signal capable of promoting developmental arrest or death[8,40]. We have recently showed that the Id3-deficiency does so through Id2-dependent effects on E proteins that involve a regulatory loop controlling the induction of Egr2 and Myc, which are required for expansion of NKγδT cells in the absence of Id3[54–56]. Together, these data indicate that Id3-deficiency influences the capacity of Vγ1.1Vδ6.3 TCR complexes to support the profound expansion of NKγδT cells.

Our finding that enforced expression of Vγ1.1Vδ6.3 TCR complexes instructs the profound expansion of NKγδT cells, even when expressed outside of the fetal/perinatal window to which their generation is normally restricted, suggests that the developmental restriction of NKγδT cell generation is controlled by the developmental timing of expression of the Vγ1.1Vδ6.3 TCR. Our analysis suggests that the restriction is centered on E protein control of the *Trav15d-1-dv6d-1* gene segment, which encodes the Vδ6.3 subunit[31,47]. Indeed, E proteins E2A and HEB both bind strongly near multiple *Trav15* elements and the importance of this binding was revealed by the impairment of Vγ1.1Vδ6.3 NKγδT cell development upon disruption of the E-box(es) downstream of the *Trav15d-1-dv6d-1* RSS. This finding indicates that E proteins are regulating *Trav15* V gene element utilization directly through cis-acting regulatory elements, but this does not provide an explanation for the strong preference for the *Trav15d-1-dv6d-1* element. There are two potential mechanisms by which E protein action at the *Trav15* family elements could support the preferential utilization of *Trav15d-1-dv6d-1* in supporting the development of Vγ1.1Vδ6.3 NKγδT cells. First, E proteins could selectively

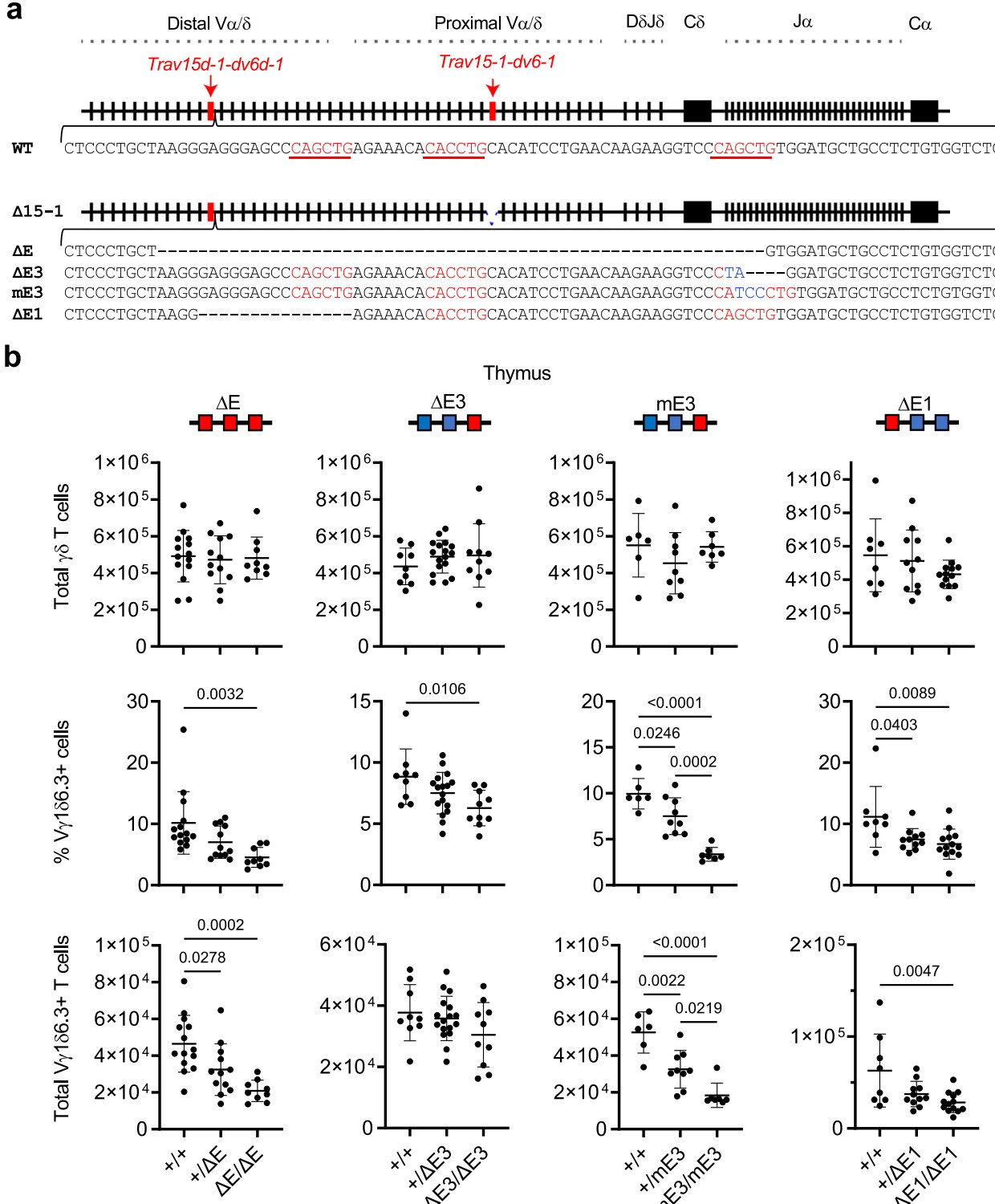

**Fig. 6 | Role of E protein binding in selection of *Trav15* family V genes during NKγδT cell development. a** Diagram of *Tcra-Tcrd* locus in 129 strain mice with relative positions of *Trav15d-1-dv6d-1* and *Trav15-1-dv6-1* indicated. The region containing E-boxes adjacent to *Trav15d-1-dv6d-1* RSS was mutated on the *Δ15-1* allele to prevent compensatory usage of the *Trav15-1-dv6-1* element. Mutations to the Ebox binding sites are indicated. **b** The number of thymic TCRδ+, and frequency and number of thymic Vγ1.1+Vδ6.3+ cells for each of the E box mutant mice (*ΔE, ΔE3,*

*mE3, and ΔE1*) are depicted graphically as scatter plots. All comparisons represent littermates. *ΔE: Tcra+/+* (*n* = 14), *Tcra+/ΔE* (*n* = 12), and *TcraΔE/ΔE* (*n* = 9); *ΔE3: Tcra+/+* (*n* = 9), *Tcra+/ΔE3* (*n* = 17), and *TcraΔE3/ΔE3* (*n* = 10); mE3: *Tcra+/+* (*n* = 6), *Tcra+/mE3* (*n* = 9), and *TcramE3/mE3* (*n* = 7); *ΔE1: Tcra+/+* (*n* = 8), *Tcra+/ΔE1* (*n* = 11), and *TcraΔE1/ΔE1* (*n* = 13). Data were pooled from at least 3 independent experiments and are plotted as mean ± SD. Statistical analysis: one-way ANOVA with correction for multiple comparison using Tukey's post hoc test.

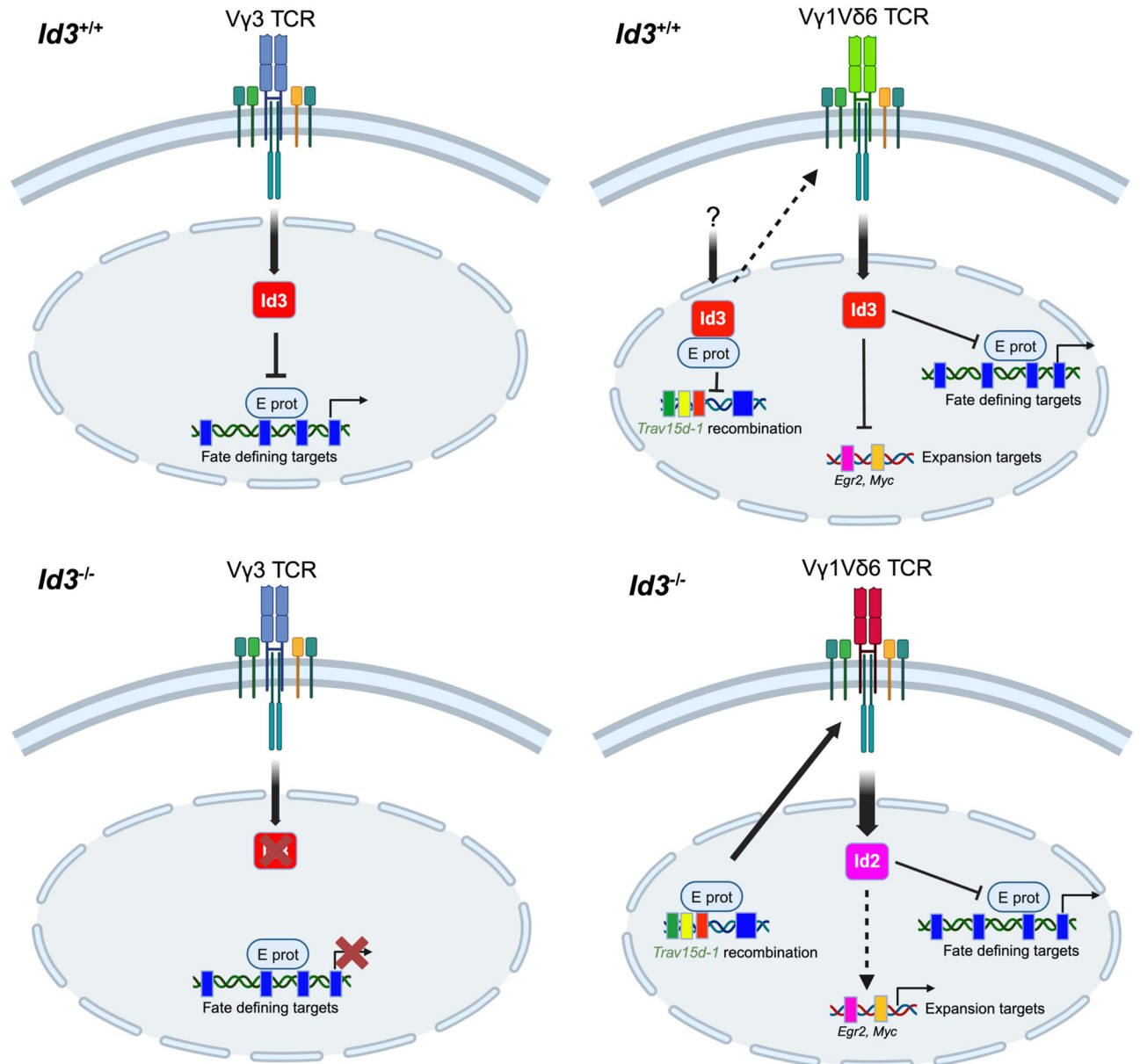

**Fig. 7 | Model of Id3 regulation of NKγδT cell development.** Separation of the αβ and γδ T cell lineages is dependent upon differences in TCR signal strength, with strong signals promoting the γδ T cell fate through suppression of E protein DNA binding through induction of the E protein antagonist Id3. Consistent with this model, development of most γδ T cell populations (e.g., Vγ3+ and KN6 γδ TCR transgenic progenitors) is attenuated when Id3 is eliminated through genetic ablation because their development requires Id3-mediated remodeling of E protein (E2A/HEB) regulation of important fate-specifying genes. In contrast, Vγ1Vδ6 expressing NKγδT cells are not suppressed, but are instead expanded. Our data indicate that this occurs because the absence of Id3 promotes the development of Vγ1Vδ6 expressing NKγδT cells in two ways. First, Id3-deficiency promotes the generation of the Vγ1Vδ6 TCR by enhancing E protein binding to the *Trav15d-1* V gene that encodes the Vδ6 subunit. Second, in the absence of Id3, the stronger signals transduced by a selected subset (red) of Vγ1Vδ6 TCR complexes with higher affinity for ligand coopts Id2, which is capable of compensating for Id3 in modulating key γδ fate specifying gene targets. Id2 also contributes to the expression of key gene targets like *Myc* and *Egr2*, upon which NKγδT cell expansion depends. This figure was produced using BioRender using our full institutional license.

promote V(D)J recombination of particular *Trav15* family members through preferential binding; however, our E protein ChIP-seq analysis revealed strong E protein binding to the RSS sequences adjacent to both *Trav15d-1-dv6d-1* and *Trav15-1-dv6-1*, which is inconsistent with this notion. Moreover, our deletion analysis clearly showed that the disfavored *Trav15-1-dv6-1* gene segment could only support development of NKγδT cells if both the preferred *Trav15d-1-dv6d-1* gene segment and Id3 were ablated. Finally, NKγδT thymocytes have been reported to contain non-functional *Tcrd* rearrangements utilizing a variety of Vδ gene segments, indicating their precursors are capable of rearranging any Vδ gene segment, but that only particular Vδ gene segments can support NKγδT development[47]. The second possibility is that the TCRδ subunit resulting from rearrangement of *Trav15d-1-dv6d-1* confers a competitive advantage to the resulting Vγ1.1Vδ6.3 TCR expressing progenitors. Interestingly, the CDR1, CDR2, and adjacent sequences encoded by the *Trav15d-1-dv6d-1* and *Trav15-1-dv6-1* elements differ by 10 amino acids (aa), including several charged aa and a bulky proline residue. These charged residues might impact the extent of interaction with polyanionic surfaces, which have been implicated in activation of Vγ1.1Vδ6.3 TCRs[53]. Accordingly, it is possible that the *Trav15d-1-dv6d-1* and *Trav15-1-dv6-1* elements are rearranged with the same frequency, but that the ten aa that differ between the

Vδ6.3 and Vδ6.1 subunits confer a selective advantage, perhaps in response to ligand engagement. Similarly, germline encoded motifs in mouse and human Vγ subunits have been implicated in promoting γδ T cell activation[57,58] and development through interaction with butyrophilin-like ligands[18,59–61]. The importance of these 10 aa differences in development and the putative ligand with which they might interact is under investigation.

The signal strength model stipulates that the induction of Id3 in proportion to TCR signal strength results in graded reductions in E protein activity that play a critical role in separation of the αβ and γδ lineages. While there is significant data in support for this model and the role played by Id3 in supporting these fate decisions, Vγ1.1Vδ6.3 expressing NKγδT cells represent an exception, since their development does not require Id3. On the contrary, their development is restrained by Id3. We provide here clear evidence that Id3 plays a clear role in both generating the stereotyped Vγ1.1Vδ6.3 TCR that promotes NKγδT cell development and its capacity to drive the marked expansion of NKγδT cells in the context of Id3-deficiency. Importantly, the Vγ1.1Vδ6.3 TCR can instruct the NKγδT fate in a cell autonomous manner at any developmental window where it is expressed, indicating the developmental restriction of this lineage is regulated at the level of Vγ1.1Vδ6.3 TCR generation. Greater insight into the relative competitive fitness of particular Vγ1.1Vδ6.3 TCR complexes in supporting NKγδT development in the absence of Id3 will ultimately depend on identifying the putative selecting ligand(s).

## Methods

### Mice
Generation of the $Tcrd^{CreER}$, $Id3^{-/-}$, $Id3^{fl/fl}$, and $Lck^{Cre}$ lines was described previously[45,62–64]. All mice were maintained in conventional housing in Association for Assessment and Accreditation of Laboratory Animal Care (AAALAC)-accredited laboratory animal facilities at either Fox Chase Cancer Center or Duke University. All experiments were conducted using littermates under protocols (02-11) approved by the institutional animal care and use committee (IACUC). Unless otherwise specified, animals employed were C57BL/6 background and were 6-8 weeks old. Equal numbers of male and female animals were employed and no association of phenotype with sex was detected. Euthanasia was performed using $CO_2$ asphyxiation.

### Tissue isolation and flow cytometry
Flow cytometry was performed on single cell suspensions from thymus, spleen and epidermis as described[65]. Cells were isolated and stained with the following antibodies. Unless stated otherwise, all antibodies were purchased from Biolegend and used at a dilution of 1:50: anti-CD3 (17A2;cat#100273), anti-CD90.2 (30-H12;cat#140311), anti-CD4 (GK1.5;cat#100453), anti-CD8 (53-6.7;cat#100752), anti-CD24 (M1/69;cat#101827), anti-CD73 (TY/11.8;cat#127223), anti-CD122 (TM-b1;cat#123207), anti-TCRβ (H57-597;cat#109222), anti-TCRδ (GL3; Invitrogen cat#15-5711-82), anti-Vγ1 (2.11;cat#141107), anti-Vγ2 (UC3-10A6;cat#137703), anti-Vγ3 (536;cat#137503), anti-Vδ6.3 (C504.17 C; BD Biosciences cat#744472), anti-PLZF (9E12;cat#145803), anti-NK1.1 (PK136;cat#108707), anti-IL4 (11B11;cat#504118) and anti-IFNγ (XMG1.2;cat#505825). Conventional flow cytometry was performed as described using DAPI or propidium iodide (PI) to exclude dead cells[48]. Intracellular flow cytometry for cytokine production was performed as previously described[48]. Cytokine production was assessed by intracellular staining of splenocytes using anti-IL4 and anti-IFNγ antibodies after stimulation for 30 min with 50 ng/ml Phorbol 12-myristate 13-acetate (PMA) and 1 μg/ml Ionomycin followed by culture for another 5 h and 30 min with 1 μg Brefeldin A. Data were analyzed on either an LSRII, Fortessa X20 or Symphony A5 flow cytometer (BD Biosciences). FlowJo Software (TreeStar) was used to analyze data. Cell populations were purified by flow cytometry using a FACSAria Cell Sorter (BD Biosciences). Gating schema for all experiments are depicted in Supplementary Figs. 5, 6 and 7.

### Fetal thymic organ culture (FTOC)
Following timed mating of mice, fetal thymic lobes were isolated from mice at E15.5, following which they were cultured in vitro at the air-medium interface, as described[6]. Fetal thymic lobes were cultured on filter discs on gelfoam at the air-medium interface in Iscove's modified Dulbecco's medium supplemented with 10 mM HEPES buffer, non-essential amino acids, 4 mM l-glutamine, penicillin, streptomycin (all from Gibco-BRL), $5 \times 10^{-5}$ M β-mercaptoethanol, and 20% fetal calf serum. Developmental progression was monitored daily by flow cytometry on quadruplicate thymic lobes.

### Single cell TCR sequencing
Individual CD24$^{hi}$ or CD24$^{low}$ Vγ1.1$^+$ γδ T cells were isolated from $Id3^{+/+}$ and $Id3^{-/-}$ mice by flow cytometry and deposited in wells of glass slides (Beckman-Coulter). Vγ1.1 and Vδ6.3 subunits were then amplified by RT-PCR using nested primers in single cells: Vγ1 external F-gggcttgggcagctggagca; Cγ4 R-gaaggaaggaaaatagtagg; Vγ1 internal F-agtatctaatatatgtctca; Cγ4 R-ggagaaaagtctgagtcagt; Vδ6 external F-ggatctaatgtggccgaga; Cδ R-tgttccatttttcatgatga; Vδ6 internal F-aagt-gattcaggtctggtca; Cδ R-tggtttggccggaggctggc. Following amplification, the cDNA fragments were analyzed by Sanger sequencing to assess the impact of Id3-deficiency on the CDR3 sequences. Following isolation, the Vγ1.1 and Vδ6.3 pairs were cloned into pMiCherry as Tescovirus 2A-linked Vγ1.1Vδ6.3 fusion proteins, as described[8].

### Structural modeling of the Vγ1.1Vδ6.3 TCR complexes
Structure predictions were performed with AlphaFold-Multimer v2.3 (https://www.biorxiv.org/content/10.1101/2021.10.04.463034v2.abstract) as implemented in ColabFold running on local machines[66]. The structures of each γδ TCR were predicted with 20 random seeds without templates with all five AlphaFold-Multimer models, for a total of 100 predictions with each. Models were relaxed with AMBER within ColabFold. Models with pTM scores lower than 0.8 were excluded, since they involved unphysiological domain-swapped dimers and other anomalies. Figures were created within PyMol. Sequence logos were produced with WebLogo[67]. Sequences for each group (KO vs WT × CD24$^{hi}$ vs CD24$^{low}$) were aligned manually in a text editor. Sequences were aligned to the longest sequences such that the length of the sequence logo is the length of the longest sequence. Identical sequences were not pruned before producing the sequence logo.

### Generation of mutant mouse lines
A Vγ1.1Vδ6.3 TCR complex that exhibited expanded representation among CD24$^{low}$ mature Vγ1.1Vδ6.3 expressing cells from Id3-deficient mice (KO TCR) and a Vγ1.1Vδ6.3 complex from Id3-sufficient mice that was not expanded in Id3-deficient mice (WT TCR), were selected for production of inducible TCR Tg mice. The WT and KO 2A-linked Vγ1.1Vδ6.3 TCR constructs that were preceded by a lox-stop-lox (LSL) sequence were cloned between the short and long-arm of the $Rosa26$ locus in the Ai6 targeting vector. G4 mouse embryonic stem (ES) cells were then transfected with the Ai6 targeting vector to knock the TCR constructs into the $Rosa26$ locus. The targeted ES cells were injected into blastocytes by the Duke University Cancer Institute Transgenic and Knockout Mouse Shared Resource and the resulting chimeric mice were screened for transmission of the targeted allele to offspring. The $Trav15d$-$1$-$dv6d$-$1$ and $Trav15$-$1$-$dv6$-$1$ gene segment deletions, and the $Tcra$-$Tcrd$ locus 430 kb deletion ($Tcra^{Δ430}$ allele) were generated using the same CRISPR/Cas9 targeting strategy reported previously[68]. Briefly, the deletions were generated using two gRNA flanking the element to be targeted (5′-TCTTCCCTTAAAGAGTGATA-3′ and 5′-GACATTAGAGTCCCTTAAAG-3′) by CRISPR/Cas9 electroporation into embryos from $Tcrd^{CreER/CreER};Id3^{fl/fl};R26^{ZsG/ZsG}$ mice on a C57BL/6 strain

background[45,63,69]. Founders were screened by Sanger sequencing and maintained separately by crossing to *Tcrd*[CreER/CreER];*Id3*[fl/fl];*R26*[ZsG/ZsG]. Deletions of the E-box containing region downstream of *Trav15d-1-dv6d-1* were similarly generated in *Tcrd*[Δ1S-1/Δ1S-1];*Id3*[fl/fl];*R26*[ZsG/ZsG] mice using two-guides flanking the target sequences to be ablated (5′-TGC AGGTGTGTTTCTCAGCT-3′ and 5′-GAACAAGAAGGTCCCAGCTG-3′). Founders were screened by Sanger sequencing and maintained separately by crossing to *Tcrd*[CreER/CreER];*Id3*[fl/fl];*R26*[ZsG/ZsG].

### Tamoxifen induction of the Vγ1.1Vδ6.3 TCR Tg

WT and KO Vγ1.1Vδ6.3 Tg on Id3-sufficient or Id3-deficient backgrounds were induced by tamoxifen treatment in vivo. Mice were injected i.p. with 1 mg of tamoxifen on days 1, 3, and 5, following which support of NKγδT cell development was assessed by flow cytometry on either day 16 or 17, as indicated. TCR Tg induction was monitored using a Zs-Green LSL reporter cassette. For induction of Vγ1.1Vδ6.3 Tg in vitro, either E16.5 FL or adult bone marrow were isolated from Vγ1.1Vδ6.3 Tg mice. FLs were disaggregated by manual disruption. Lineage negative hematopoietic stem and progenitor cells (HSPC) were isolated from bone marrow using magnetic bead depletion with biotinylated antibodies (B220, CD3, CD4, CD8, CD11b, CD11c, CD19, CD44, GR-1, IgM, NK1.1, TCRβ, and TCRγδ) and streptavidin magnetic beads. To evaluate the capacity of the WT and KO Vγ1.1Vδ6.3 Tg to promote adoption of the NKγδT fate in vitro, FL or bone marrow HSPC were cultured on OP9-DL1 cells in the presence of 1 ng/mL IL-7 and 5 ng/mL FLT3L (BioLegend) as previously described[70]. After 8 days in polarizing conditions, cells were placed in experimental conditions. Tg TCRs were induced using 4-hydroxytamoxifen (250 ng/mL). Cells were cultured on OP9-DL1 cells in the presence of 1 ng/mL IL-7 and 5 ng/mL FLT3L until day 6 post-induction, and only in the presence of 1 ng/mL IL-7 thereafter (BioLegend). Cultures were analyzed and passaged every 3 days.

### ChIP-seq analysis

Chip-Seq for E2A and HEB was performed on DN3 cell generated from *Rag2*[−/−] FL progenitors by culture for 7 days in IL-7 and Flt3 ligand on OP9-DL1 monolayers[48]. The resulting ChIP-seq analysis was performed as described on formaldehyde fixed and sonicated chromatin and has been deposited in GEO (GEO: GSE162292)[48]. The resulting data were reanalyzed here. Alignment was performed using Bowtie2 (multiple read alignment, $k = 3$), peak calling was performed using MACS2 (default options were used), following which data were visualized using Integrative Genomics Viewer. Motif analysis was performed with Homer using the findMotifsGenome.pl function (size = 200, masked).

### Statistical analysis

Sample size was not predetermined by statistical methods. Statistical significance was determined using GraphPad Prism, which was also used for generation of graphs, with the statistical test indicated on the legend and *p* values listed on all graphs.

### Reporting summary

Further information on research design is available in the Nature Portfolio Reporting Summary linked to this article.

## Data availability

The data in the figures are available in the published article or its supplementary files. Unprocessed FCS files and processed FlowJo files are available at the FlowLIMS (http://sysbio.fccc.edu:8087/lims/Login.jsp) at Fox Chase Cancer Center (FCCC) by request. Access is restricted due to Temple University Health System privacy restrictions. The ChIP-Seq data used in this study are available from the Gene Expression Omnibus under accession number GSE162292. Source data are provided with this paper.

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

## Acknowledgements

Finally, we gratefully acknowledge the assistance of the following core facilities of the Fox Chase Cancer Center: Cell Culture, Flow Cytometry, Laboratory Animal, and Molecular Modeling. We also gratefully acknowledge the assistance of the Flow Cytometry Shared Resource and the Transgenic and Knockout Mouse Shared Resource of the Duke University Cancer Institute, and the Duke University Division of Laboratory Animal Resources. D.L.W. was supported by NIH grant P01AI102853, core grant P30CA006927, the Bishop Fund and an appropriation from the Commonwealth of Pennsylvania. R.L.D. was supported by R35GM122517, Y.Z. was supported by NIH grants P01AI102853 and R01GM059638, and M.C. was supported by P01AI102853 and R01GM115474. The authors declare no competing financial interests.

## Author contributions

D.L.W., Y.Z. and M.C. conceived and oversaw the study and together with A.M. wrote the manuscript. A.M., S.Y.L., S.S., A.V.C., B.Z., M.R., M.I.P. and R.D. performed experiments and analyzed data. J.C.Z.P. contributed critical reagents.

## Competing interests

The authors declare no competing interests.
