## [Peer Review File · Nature Communications]

REVIEWER COMMENTS

Reviewer #1 (Remarks to the Author):

The manuscript by Wiest and colleagues provides interesting new data revealing that E proteins control the development of mouse NKgdT cells by directly regulating the T cell receptor (TCR) expressed by these cells. The CHIP-seq analysis and the generation of mice lacking specific E protein binding sites adjacent to the NKgdT TCR was very convincing and overall, the manuscript was well written. However, some aspects of the science need to be addressed prior to publication and I am concerned about the quality of parts of this research. I detail my concerns below:

1. Sample size should always be included for all experiments. This has been done in the later figures but is absent in Figures 1 & 2. Please justify the use of a t- test.
2. Figures 1 & 2. Representative data should also be shown for Id3+/- mice and percentages (not just total cell numbers) should be graphed for these experiments.
3. Figure 1c. Why is the background staining so different for V α 1.1 between Id3+/+ and Id3-/- mice. Are V α 1.1+TCR α - cells real? The gate for V α 1.1+TCR α + cells in Id3-/- mice doesn't appear to contain all positive cells. These data are a concern.
4. There is no discussion of Figure 3c and the inclusion of this figure probably does not add to the manuscript.
5. Figure 4, extended Figure 3 and extended figure 4. The data showing differences between TCRs isolated from Id3+/+ and Id3-/- mice in their ability to promote adoption of the NKgdT cell fate is not convincing for several reasons. There are differences in TCR α expression in 1c suggesting these representative experiments were performed on different days. There is a compensation issue with PLZF versus V α 1.1. In some cases, the representative data does not look representative. For example, the graphed mean for KO TCR into Id3-/- is ~60% and WT TCR into Id3-/- ~45%, whereas the representative numbers are much higher. PLZF gating is inaccurate, specifically KO TCR into Id3-/- mice which cuts the population of PLZF- negative cells in half. The levels of significance shown for 2-3 fold differences appears inflated, i.e. ****. Similar 2-3 fold differences observed in similar experiments in extended figure 3 do not reach the same level of significance, often just one *. Please check to ensure the accuracy of these data. Similar, in extended figure 4b, how can significance of 4 stars be reached with such large error bars. Sample size is needed for extended figure 4b.
6. X and Y labels are missing from extended figure 5b.
7. All flow plots should include numbers on axis.
8. Do the authors think that other innate like T cell TCRs are regulated by E proteins? NKT cells, MAIT cells?

Reviewer #2 (Remarks to the Author):

E proteins control NK γ δ T development through both generation and function of their stereotypic TCR

Previous studies indicated that NK γ δ T cells that express V γ 1.1V δ 6.3 show substantial expansion in mice that are deficient for Id3. While interesting these findings raised a number of questions that have puzzled the field for quite some time. Here the authors explore the mechanism that underpins by which a deficiency a deficiency of Id3-expression leads to expanding V γ 1.1V δ 6.3 NKgd T cells. The authors find that E-proteins and Id3 regulate Vd6.3 gene expression. Furthermore, the Vg1Vd6.3 expressing cells that expand in Id3-deficient mice in a cell-autonomous and developmentally-unrestricted manner, also readily adopt an NKgdT cell fate. These data provide new insight for the substantial expansion of Vg1Vd6.3 in Id3-deficient mice. In sum, these observations indicate that the ability of NKgdT cells to markedly expand in the absence of Id3 expression is dictated by the expression of the Vg1Vd6.3 TCR complex.

In my view this is an interesting study. It provides substantial new insights into mechanisms that instruct gdT cell development. The data provide rather compelling evidence, using a series of well-designed and clever genetic manipulation, that the expansion of NK gdT cells in the absence of Id3 expression is regulated by the expression of the Vg1Vd3 TCR complex. Finally, this problem in gd-lineage development appears to be solved! The manuscript is also well written and the data are presented in a logical and coherent manner.

Comments:

1. Please include plots indicating percentages of the various populations (Figure 1).
2. Please include cell numbers and p-values when comparing E175. Id3 WT versus Id3 MUT fetal thymocytes (Figure 2a).
3. Could the authors please include a motif analysis for the ChIP-seq analysis to confirm that indeed E-box sites are highly enriched using both E2A and HEB antibodies?
4. A model is needed demonstrating the activities of E- and Id-proteins controlling the expression of genes that are targeted and potential feed-back loops involving E2A/HEB and Id3.

5. The discussion section needs to be shortened.

6. Spelling error- “once expressed, the resulting Vg1Vd6.3 TCR in capable” should be- is capable.

7. Spelling error “co-producton of IL-4 and IFN γ without the need for prior activation” should be co-production.

Mihai et al., response to review

We thank the reviewers for their encouraging and constructive comments as addressing them has strengthened the manuscript.

Reviewer 1 concluded that our manuscript comprised interesting new data revealing that E proteins control the development of mouse NKgdT cells by directly regulating the T cell receptor (TCR) expressed by these cells. However, it was felt that some aspects of the science needed to be addressed prior to publication.

1. Sample size should always be included for all experiments. This has been done in the later figures but is absent in Figures 1 & 2. Please justify the use of a t- test.

We have gone through the data and provided all of the requested information in the figure legends. We initially used the t test to assess statistical significance; however, after consultation with our Biostatistics Facility, it was decided that one-way Anova and two-way Anova with correction for multiple comparisons using Tukey's post hoc testing was more appropriate for Figures 1 and 2, respectively. None of the conclusions were impacted.

2. Figures 1 & 2. Representative data should also be shown for Id3+/- mice and percentages (not just total cell numbers) should be graphed for these experiments.

The requested data elements have been provided.

3. Figure 1c. Why is the background staining so different for V γ 1.1 between Id3+/+ and Id3-/- mice. Are V γ 1.1+TCR δ - cells real? The gate for V γ 1.1+TCR δ + cells in Id3-/- mice doesn't appear to contain all positive cells. These data are a concern.

We have re-evaluated the data and found some variability in the background staining of the TCR δ negative cells but the V γ 1/TCR δ positive cells stain consistently and are clearly distinct from background.

4. There is no discussion of Figure 3c and the inclusion of this figure probably does not add to the manuscript.

The reviewer is correct that it is not essential, but does make the point that the CDR3s from the Id3-/- TCRs have less flexibility and freedom of movement. We have updated the text to briefly mention this.

5. Figure 4, extended Figure 3 and extended figure 4. The data showing differences between TCRs isolated from Id3+/+ and Id3-/- mice in their ability to promote adoption of the NKgdT cell fate is not convincing for several reasons. There are differences in TCR δ expression in 1c suggesting these representative experiments were performed on different days. There is a compensation issue with PLZF versus V γ 1.1. In some cases, the representative data does not look representative. For example, the graphed mean for KO TCR into Id3-/- is ~60% and WT TCR into Id3-/- ~45%, whereas the representative numbers are much higher. PLZF gating is inaccurate, specifically KO TCR into Id3-/- mice which cuts the population of PLZF- negative cells in half. The levels of significance shown for 2-3 fold differences appears inflated, i.e. ****. Similar 2-3 fold differences observed in similar experiments in extended figure 3 do not reach the same level of significance, often just one *. Please check to ensure the accuracy of these data. Similar, in extended figure 4b, how can significance of 4 stars be reached with such large error bars. Sample size is needed for extended figure 4b.

We have re-examined all of the data, adjusted the gating to ensure its accuracy and consistency, and have reassessed the statistical significance of our findings using two-way Anova. The p-values are now listed on the figures and the conclusions were not affected. Finally, we have changed the figure to ensure that representative profiles are provided that reflect the mean values in the different populations.

6. X and Y labels are missing from extended figure 5b.

The labels have been added to what is now Extended Figure 5c.

7. All flow plots should include numbers on axis.

Decade numbers has been added to all flow plots.

8. Do the authors think that other innate like T cell TCRs are regulated by E proteins? NKT cells, MAIT cells?

The reviewer raises an interesting point. It is possible that E proteins play an important role in regulating other innate T cell lineages. For example, we have recently examined the impact of Id3-deficiency on innate CD8 T cells

and found them to be markedly increased in the thymus; however, the basis for this expansion has not yet been established and is the subject of ongoing analysis.

Reviewer 2 felt that our study provides rather compelling evidence, using a series of well-designed and clever genetic manipulation, that the expansion of NK gdT cells in the absence of Id3 expression is regulated by the expression of the Vg1Vd3 TCR complex and that finally, this problem in gd-lineage development appears to be solved! The reviewer also felt the manuscript was well written and the data are presented in a logical and coherent manner. We thank reviewer 2 for those kind words.

Comments:

1. Please include plots indicating percentages of the various populations (Figure 1).

Plots and graphs of population frequencies have been added to Figures 1 and 2.

2. Please include cell numbers and p-values when comparing E175. Id3 WT versus Id3 MUT fetal thymocytes (Figure 2a).

Statistical analysis for absolute numbers and population frequencies have been provided for Figure 2.

3. Could the authors please include a motif analysis for the CHIP-seq analysis to confirm that indeed E-box sites are highly enriched using both E2A and HEB antibodies?

Motif analysis for E2A and HEB has been added to Extended Figure 5a.

4. A model is needed demonstrating the activities of E- and Id-proteins controlling the expression of genes that are targeted and potential feed-back loops involving E2A/HEB and Id3.

A model has now been added to describe our understanding of the role of the Id3/E protein axis in regulating NK γ δ T cell development (Figure 7).

5. The discussion section needs to be shortened.

The discussion has been edited to make it more concise.

6. Spelling error- "once expressed, the resulting Vg1Vd6.3 TCR in capable" should be- is capable.

This has been corrected.

7. Spelling error "co-producton of IL-4 and IFN γ without the need for prior activation" should be co-production.

This has been corrected.

REVIEWERS' COMMENTS

Reviewer #1 (Remarks to the Author):

The manuscript is most improved, the authors have addressed my concerns and the changes to the figures are a big improvement to the overall quality of the data presented. Extended figure 3C contains an old label that should be covered.

Reviewer #2 (Remarks to the Author):

The authors have addressed the vast majority of questions. It is now in excellent shape. The manuscript is now suitable for publication in Nature Communications.

Mihai et al., response to review

We thank the reviewers for their encouraging and constructive comments as addressing them has strengthened the manuscript.

Reviewer 1 noted that original extended Figure 3C had an old label that needed to be covered up.

This has been completed for what is now Figure S2.